# Cryo-EM captures early ribosome assembly in action

Bo Qin [1,8], Simon M. Lauer [1,8], Annika Balke [2], Carlos H. Vieira-Vieira [3,4], Jörg Bürger [1,5], Thorsten Mielke [5], Matthias Selbach [3,6], Patrick Scheerer [2], Christian M. T. Spahn [1] ✉ & Rainer Nikolay [1,7] ✉

Ribosome biogenesis is a fundamental multi-step cellular process in all domains of life that involves the production, processing, folding, and modification of ribosomal RNAs (rRNAs) and ribosomal proteins. To obtain insights into the still unexplored early assembly phase of the bacterial 50S subunit, we exploited a minimal in vitro reconstitution system using purified ribosomal components and scalable reaction conditions. Time-limited assembly assays combined with cryo-EM analysis visualizes the structurally complex assembly pathway starting with a particle consisting of ordered density for only ~500 nucleotides of 23S rRNA domain I and three ribosomal proteins. In addition, our structural analysis reveals that early 50S assembly occurs in a domain-wise fashion, while late 50S assembly proceeds incrementally. Furthermore, we find that both ribosomal proteins and folded rRNA helices, occupying surface exposed regions on pre-50S particles, induce, or stabilize rRNA folds within adjacent regions, thereby creating cooperativity.

The assembly of ribosomal subunits, the largest ribonucleoprotein particles in the cell, relies on the intrinsic affinities of their rRNA and protein components that interact in a hierarchical manner[1,2], involving protein-induced conformational changes in rRNA[3]. The resulting cooperativity of assembly ensures the completion of ribosomal subunits[4]. The *E. coli* 50S large ribosomal subunit is composed of 33 proteins (L-proteins), one 23S and one 5S rRNA. The 23S rRNA consists of six architectural domains that, stabilized by the L-proteins, define the shape of the 50S subunit with its three protuberances[5–7]. Cryo-EM studies of bacterial 50S precursors both purified from cells[8–13] and obtained by in vitro assembly[14] have elucidated the later phase of the maturation pathway.

The most immature 50S precursor states reported contained an already fully formed core, consisting of 23S rRNA domains I, II, III, VI and early binding L-proteins[10,12,14]. In this core particle about 50% of all protein and RNA residues are already maturely positioned[14]. Further assembly involves the formation of the L1 stalk, the central protuberance (CP), the GTPase associated center (GAC), and the stalk base. 50S assembly terminates with folding of the peptidyl transferase center (PTC) and the surrounding rRNA elements referred to as functional core (FC)[8–10,12,14–16]. The events after structural formation of the large subunit´s core appear to be evolutionary conserved. Hence, similar assembly intermediates have been observed for the eukaryotic 60S subunit[17–19] and large subunit precursors derived from mitochondria, which shed light on the latest period of assembly, detailing the finely tuned steps leading to completion of the subunit´s active site[20–25], although the involved biogenesis factors differ substantially[13].

[1]Institut für Medizinische Physik und Biophysik, Charité – Universitätsmedizin Berlin, corporate member of Freie Universität Berlin and Humboldt Universität zu Berlin, Berlin, Germany. [2]Charité – Universitätsmedizin Berlin, corporate member of Freie Universität Berlin and Humboldt-Universität zu Berlin, Institute of Medical Physics and Biophysics, Group Protein X-ray Crystallography and Signal Transduction, Charitéplatz 1, D-10117 Berlin, Germany. [3]Proteome Dynamics, Max Delbrück Center for Molecular Medicine in the Helmholtz Association (MDC), Robert-Rössle-Str. 10, 13125 Berlin, Germany. [4]Faculty of Life Sciences, Humboldt Universität zu Berlin, Berlin, Germany. [5]Microscopy and Cryo-Electron Microscopy Service Group, Max Planck Institute for Molecular Genetics, Ihnestr. 63-73, 14195 Berlin, Germany. [6]Charité -Universitätsmedizin Berlin, 10117 Berlin, Germany. [7]Department of Genome Regulation, Max Planck Institute for Molecular Genetics, Ihnestr. 63-73, 14195 Berlin, Germany. [8]These authors contributed equally: Bo Qin, Simon M. Lauer. ✉e-mail: christian.spahn@charite.de; nikolay@molgen.mpg.de

In comparison, early assembly of the large ribosomal subunit remains elusive from a structural standpoint. Pioneering studies visualizing nucleolar derived samples from yeast provided first structural insight into intermediate states before core formation has been completed[26,27]. However, large subunit precursors exploring the earliest stages of their assembly escaped structural analyses so far. Here, we present a biochemical and cryo-EM based analysis of the earliest period of 50S assembly, utilizing an in vitro reconstitution system with purified rRNA and L-protein components. We provide high resolution structures of early pre-50S intermediates explaining their structural maturation via parallel assembly routes, starting with a particle consisting of density corresponding to the first ~500 nucleotides of the 23S rRNA and the proteins uL22, uL24, and uL29.

## Results

### Time course of 50S in vitro reconstitution

To focus on early events of 50S assembly, we adopted our previous approach[14] and performed step 1 of the in vitro reconstitution assay[28] as a time course reaction, analyzed sucrose gradient profiles, translation activity of the assembled particles (after a full step 2 incubation), and determined the protein occupancy, using quantitative mass spectrometry (q-MS) ("Methods", Fig. 1a and Supplementary Fig. 1a–j). Before heat incubation, the sample migrates in a sucrose gradient as a 33S precursor (Fig. 1b) with reduced amounts of the L-proteins bL32, uL30, uL14, uL29, bL19, bL28 and uL16 (Supplementary Fig. 1j), and exhibits no translation activity (Fig. 1c), in agreement with previous studies[29]. As expected, both the portion of faster migrating particles and the translation activity increased with the time of incubation (Fig. 1c and Supplementary Fig. 1b–h).

The sucrose gradient profiles revealed that after 3 min of reaction, the sample exhibited its largest complexity with the majority of precursors in the 33S region and a significant amount of faster migrating particles (Fig. 1b). Nevertheless, particles in the 3 min sample contain all L-proteins (except bL31 and bL36) with occupancies of >70% (Supplementary Fig. 1i). Hence, we selected this sample for a detailed cryo-EM analysis. After extensive sorting and multi-particle refinement, we were able to disentangle 16 different states, with nominal resolutions between 3.0 and 6.6 Å (Supplementary Fig. 2, Supplementary Fig. 3, and Supplementary Table 1). While some of the obtained states are related to the 41S-like and 48S-like particles of the late phase of assembly that we have previously obtained with a full 30 min incubation[14], the time-course approach resulted in nine states

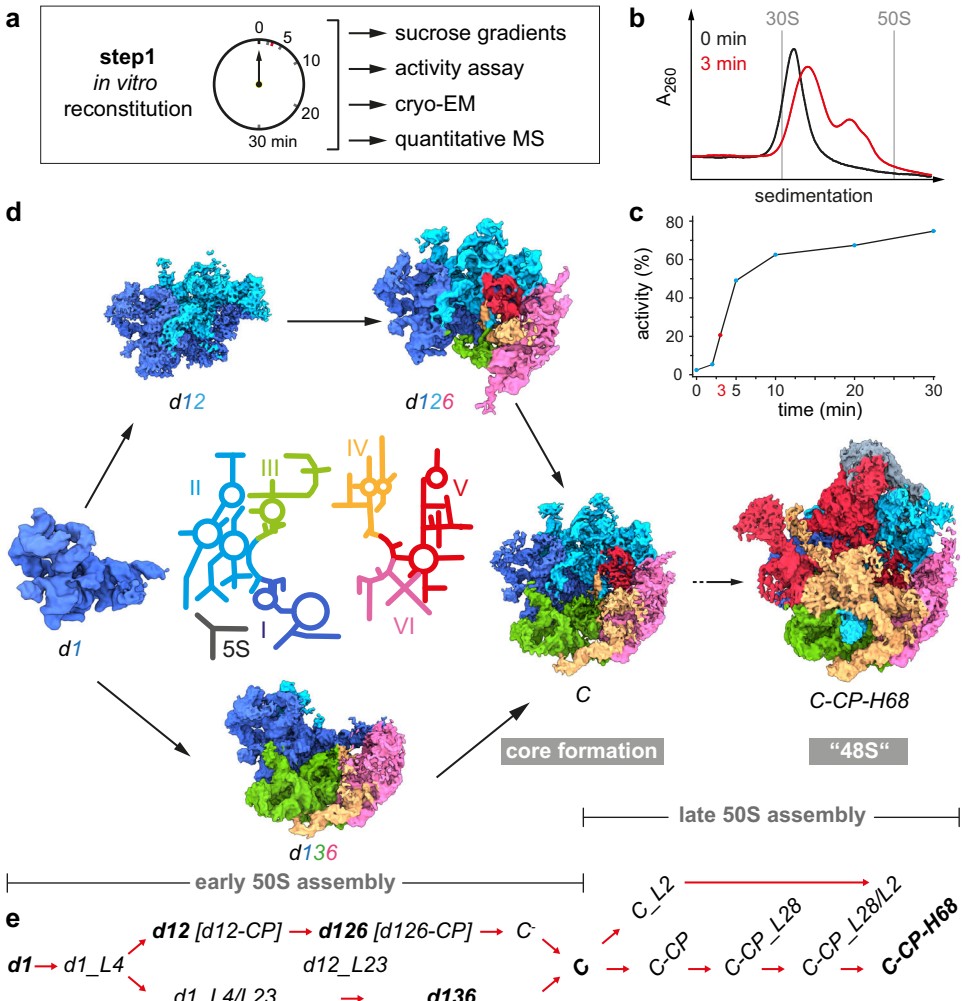

**Fig. 1 | Time course of 50S step 1 in vitro reconstitution with structural and biochemical analyses. a** Experimental setup. **b** Sucrose gradient profiles of in vitro reconstitution samples incubated for 0 min (black) or 3 min (red), respectively. **c** In vitro translation assay indicating the relative activity of subunits after the indicated time of incubation under step 1 conditions plus subsequent incubation under step 2 conditions for 90 min. Data are presented as mean values based on n = 2 technical replicates. **d** In the center, a color-coded 2D map of the 23S rRNA with the individual domains labeled from I–VI (5S, 5S rRNA). Cryo-EM maps of selected states derived from the step 1 reaction after 3 min of incubation appear in the same color-code. **e** Nomenclature for states occurring during early and late 50S assembly. States in bold are shown in (**d**). d, 23S rRNA domain; C, 50S core; CP, central protuberance; H68, helix 68. Source data are provided as a Source Data file.

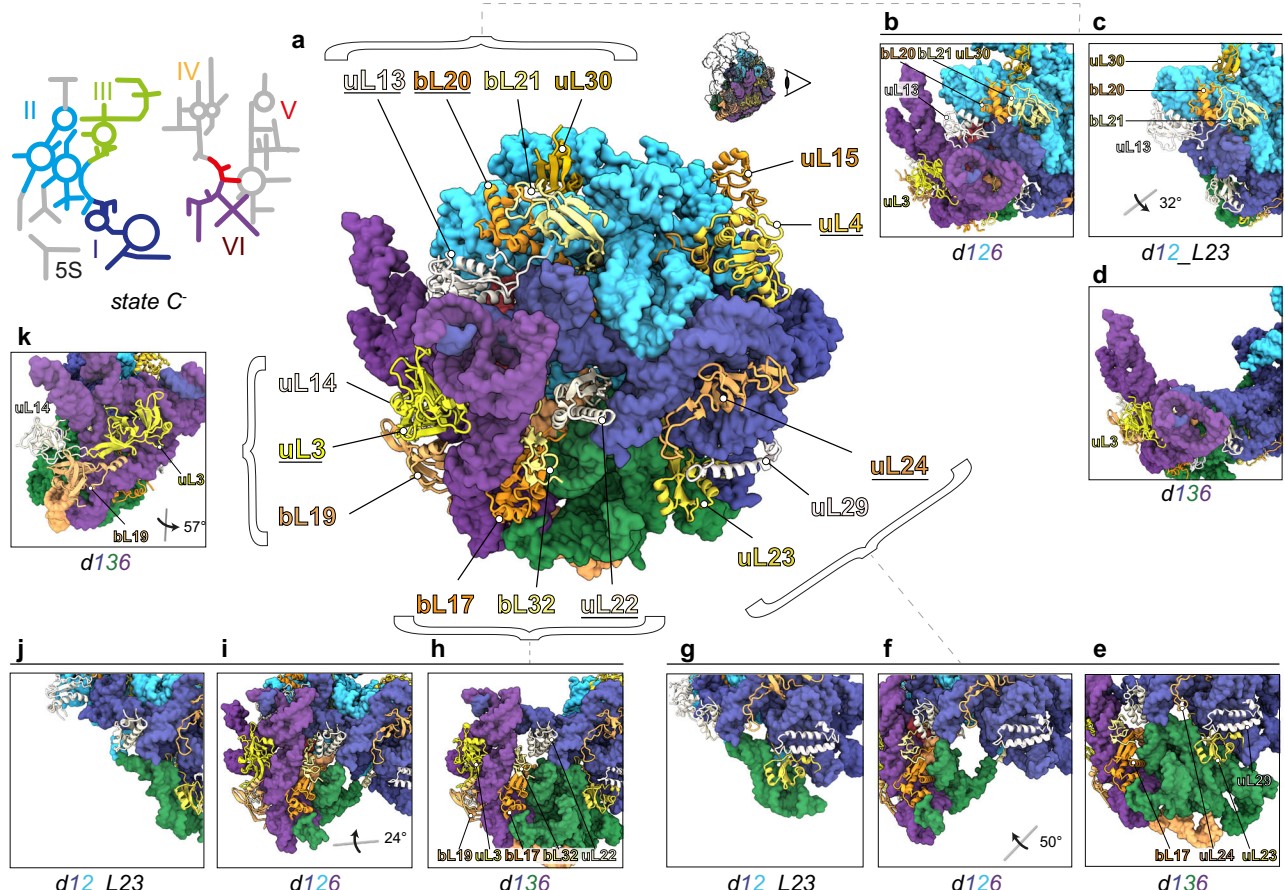

**Fig. 2 | Clustering of L-proteins. a** 2D rRNA map and atomic model of state *C* in back view. L-proteins appear as cartoons (early essential proteins and uL3 are underlined), rRNA as surface model (lowpass-filtered to 5 Å resolution). Presence and absence of bL21, bL20, and uL30 in states *d126* (**b**), *d12* (**c**), and *d136* (**d**), respectively. Presence and absence of uL23 in states *d136* (**e**), *d126* (**f**), and *d12* (**g**), respectively. Presence and absence of bL17, bL32, and uL3 and bL19 in states *d136* (**h**), *d126* (**i**), and *d12* (**j**), respectively. Positions of uL14, uL3 and bL19 in state *d136* (**k**). Viewing angles in (**c**, **f**, **i**, and **k**) are shown relative to the full model in a).

identified herein that were clearly less complete than the core particle. Whenever in the following we describe the presence, or appearance of an L-protein, an rRNA element, or an rRNA domain, it refers to a detectable, unambiguous density of this component in the corresponding cryo-EM map. Appearance of such densities can result from a component just binding at that moment or being part of the complex already, and structural rearrangements led to a more stable conformation capable of being captured by cryo-EM based technologies. This caveat is of particular importance in cases where densities for proteins and RNA elements appear simultaneously.

**Structure-based pathway for early 50S assembly**

The earliest state we identified contained strong density for ~500 bp of domain I and the L-proteins uL22, uL24, and uL29 (referred to as state *d1*, Fig. 1d and Supplementary Fig. 4a). Hence, 50S assembly in vitro starts with the stable formation of the 5´end containing domain I, despite the presence of the entire 23S rRNA molecule. Interestingly, uL24, whose binding site is in the center of *d1*, is found among the first binders (Supplementary Fig. 1i), which explains its role as an assembly initiator[29]. To better understand the molecular anatomy of the subunit, we arranged selected cryo-EM maps in a logical order and noticed that early assembly can progress at least along two putative routes (Fig. 1d).

Instead of numbering the individual states successively, we designed a more specific nomenclature (Fig. 1e). The early states were designated according to the 23S rRNA domains whose cryo-EM densities they exhibit (e.g., *d1*, *d12*, *d126*, etc.). In order to distinguish states

very similar in composition, additional structural elements were added to their names (e.g., *d1_L4/L23*, densities for domain I plus uL4 and uL23 etc.). Once densities for domains I, II, III, VI, and their interacting proteins are present, the large subunit´s core has completely formed (state *C*). To differentiate states occurring during late assembly, emerging structural features such as the central protuberance, or helix 68 were added to their names (*C-CP*, or *C-CP-H68*).

While state *d1* can mature to *d12* and *d126*, an alternative route involves state *d136*. Both states *d126* and *d136* can convert into state *C* (*d1236*), a process we refer to as core formation, that enhances base pairing between the 5' and 3' ends of 23S rRNA. Consequently, core formation in vivo can only be accomplished after transcription of the entire 23S rRNA molecule. Subsequently, state *C* transitions via intermediate steps into state *C-CP-H68* (48S) (Fig. 1d, e and Supplementary Figs. 4 and 5). Interestingly, some of the 23S rRNA domains indeed appear to be folding units that are intrinsically rigid, but flexible relative to one another (Supplementary Movie 1). Domains I and VI exhibit full cryo-EM density within their domain boundaries as soon as they become detectable as parts of discrete states (Fig. 1d; *d1*, *d126*, and *d136*). Defined regions of domains II and III show density in states *d1_L4* and *d1_L4/L23*, respectively, before subsequent concerted appearance of their remaining densities (Supplementary Fig. 4). In the case of domain II only density for the central region appears and extended parts such as A-site finger and GTPase associated center (GAC) form in late assembly. Taken together, early 50S assembly occurs in a domain-wise fashion, while late 50S assembly proceeds incrementally.

## Detailed molecular morphogenesis · Contribution of L-proteins

L-proteins concentrating on the back site of the subunit are organized in clusters, containing at least one early assembly protein, providing contact points for later binding proteins and connecting adjacent rRNA domains (Fig. 2a). To exemplify in more detail how certain L-proteins contribute to 50S assembly, we compared states *d1*, *d1_L4* and *d1_L4/L23* (Fig. 3a–e and Supplementary Movie 2). State *d1* is lacking density for uL4 (Fig. 3a), while *d1_L4* exhibits densities for uL4 along with domain II helices H27-31 (Fig. 3b). Similarly, state *d1* is lacking density for uL23 (Fig. 3d), while *d1_L4/L23* exhibits densities for uL23 along with domain III helices H49-53, including the interjacent bulges (Fig. 3e). Thus, uL4 and uL23 appear to contribute to the formation of local seeds to promote further assembly of domains II and III, respectively (Fig. 2b–j). However, while uL4 is present in all downstream assembly intermediates, presence, or absence of uL23 determines, whether assembly proceeds along route 1-3-6 or 1-2-6 (Fig. 2e–g).

Another L-protein that affects the formation of domain III is bL17, whose presence mediates stable folding of H47-49 (Fig. 2h–j). Based on these observations, we conclude that some L-proteins, occupying surface exposed regions on a pre-50S particle, are capable of folding or, stabilizing rRNA elements of adjacent domains. While it is well-known that L-proteins are frequently located at rRNA domain interfaces[6,7], and that r-proteins have the capability to remodel rRNA in a cooperative fashion[3] this dedicated function of uL4, uL23, bL17, and other L-proteins in ribosome assembly was not apparent (Supplementary Movie 3).

## Detailed molecular morphogenesis—Successive manifestation of L-proteins

Globular domains of L-proteins, like in uL24, contribute to the formation of the subunit´s crust. Some L-proteins such as uL4, uL3, bL32, and uL2, contain a globular domain (yellow) combined with extended tentacles or loops (red) that interestingly project towards the center of the subunit (Fig. 4a, b). Cryo-EM density for the globular domain of uL4 appears early after formation of domain I, while most of the loop forms with domain II, and the apical tip even later with domain V (Fig. 4c). The globular domain of uL3 critically contributes to formation of domain VI (Fig. 2h, i, k), and the extended loop forms later with domains II, IV and V (Fig. 4d). In addition, bL32 occupies a critical position with its globular domain being part of domain VI (Fig. 2a, h, i) and the extension interacting with domains I, II and V, thereby connecting four rRNA domains (Fig. 4e).

Interestingly, the globular part of uL2 is present on domain III only after core formation in state *C_L2* (Supplementary Fig. 4l) and binding can occur later in state *C-CP_L28/L2* (Supplementary Fig. 4o). Nevertheless, the extended loop starts forming earliest in state *C-CP_L28/L2* and interacts mainly with and stabilizes helices H65, H66 and H69 of domain IV. Only in a mature 50S subunit, density for the whole loop appears that in addition undergoes contacts with domain V (Fig. 4f). Taken together, the flexible tentacles of some L-proteins form later than their globular domains and interact predominantly with nucleotides of the late forming rRNA domains IV and V, possibly by mechanisms involving dynamic sampling, as shown for proteins of the 30S subunit[30,31].

## Hierarchical arrangement of rRNA

Another principle we observe is the stabilization or folding of rRNA helices, induced by adjacent rRNA elements that are nearly maturely folded. Assembly appears to nucleate at the 5´-end with the formation of domain I to yield state *d1*. To rule out biased image classification by initial alignment, we re-analyzed our cryo-EM data and aligned all particle images before proceeding with 3D classification (Supplementary Fig. 6). However, this did not result in discovery of states that would support any nucleation point other than domain I.

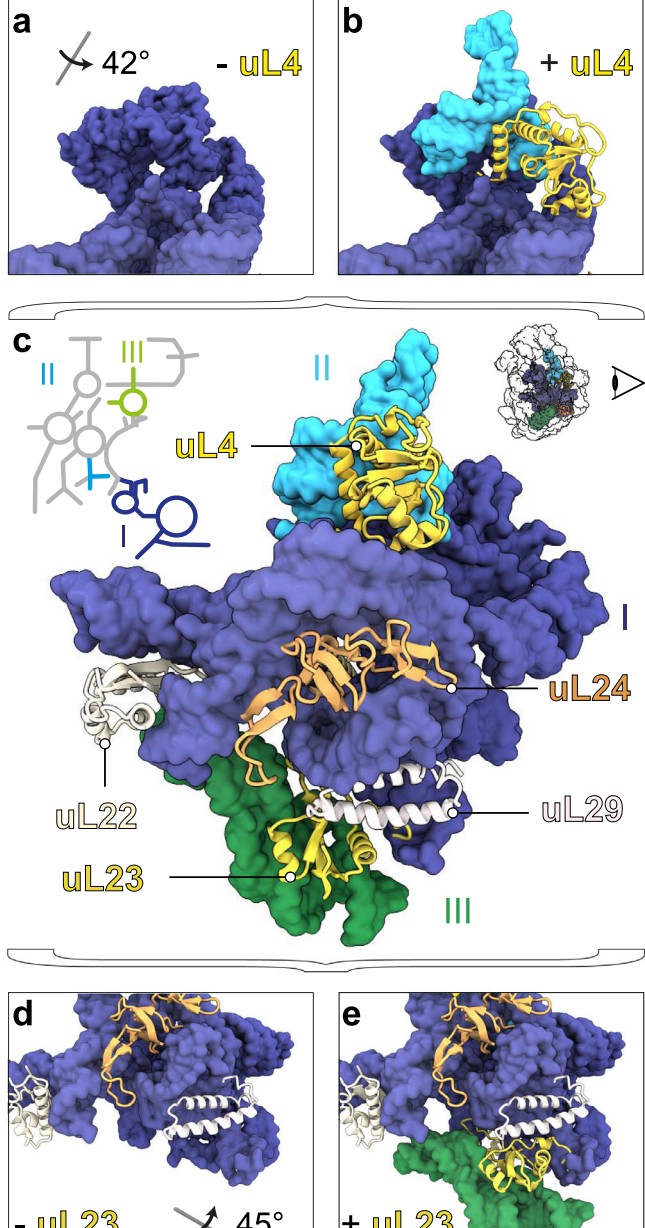

**Fig. 3 | Proteins with seeding function.** PDB models of early states with rRNA in surface representation and L-proteins as cartoons. Top segments of states *d1* (**a**) and *d1_L4* (**b**), full model of state *d1_L4/L23* in back view (**c**), and bottom segments of states *d1* (**d**) and *d1_L4/L23* (**e**) color-coded according to the 23S rRNA 2D map. All viewing angles relative to (**c**).

Thus, domain I indeed seems to appear first, while the remaining rRNA elements dock onto an already existing RNA surface.

The precursors *d12* (Fig. 5a, b) and *d136* (Fig. 5c, d) exhibit intrinsic flexibility along certain hinge regions located in domain interfaces (red dotted lines), while the individual rRNA domains remain relatively rigid. Nevertheless, the relative domain movements from the most open (Fig. 5a, c) to the close conformation (Fig. 5b, d) allow for the formation of inter-domain contacts between H11/H37 (Fig. 5b) and H21/H51 and H11/H37 (Fig. 5d). In some cases, formed helices remain flexible and are stabilized by tertiary contacts during subsequent assembly. For instance, both H21 and H22 exhibit solid density in state *C⁻*, but both helices reconfigure their shape and conformation and form tertiary contacts once reaching their mature positions (Fig. 5e–g). Further rRNA-assisted rRNA stabilization or

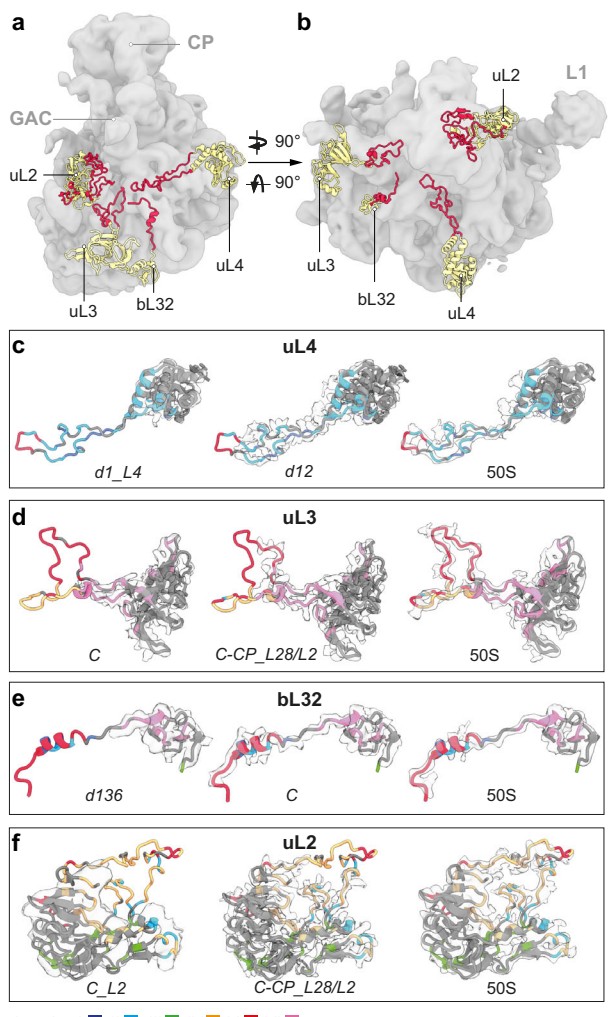

**Fig. 4 | L-proteins with extensions. a** Side view, **b** top view of a lowpass-filtered *C-CP-H68* cryo-EM map (gray) with models of selected L-proteins exhibiting globular domains (yellow) and extensions (red). CP, central protuberance; GAC, GTPase associated center; L1, L1 stalk. **c**–**f** Cryo-EM maps (light gray) of selected L-proteins, derived from the indicated states, and corresponding atomic models (EMD-22614). Protein models are colored according to their rRNA domain contacts; gray color indicates self-interaction, or surface exposure ("Methods").

folding becomes apparent, when comparing assembly along the routes 1-2-6 and 1-3-6. While route 1-2-6 initially yields a core particle lacking density for domain IV helices H61-63 (state *C⁻*), maturation along route 1-3-6 culminates in a particle with these parts of domain IV stably folded (state *C* and better defined in state *C-CP_L2/L28*) (Figs. 1d and 5h–j). Hence, initial appearance of domain IV benefits from the presence of the maturely folded domain III. Specifically, rRNA contacts with the already formed domain III helices H56-58 seem to induce formation of domain IV H61 and H63 in state *d136*, and H61-63 in state *C* (Supplementary Fig. 7a–c), thereby initiating the incremental formation of domain IV.

In addition, we noticed that the 5S RNP (ribonucleoprotein particle), consisting of 5S rRNA, uL5, uL18, and bL25, can dock on a pre-50S particle as soon as the central region of domain II has formed (Supplementary Fig. 8a, b). This can occur both in early (*d12-CP*, *d126-CP*) and late assembly (*C-CP*) and goes along with formation of domain II helices H38 (A-site finger) and H42 (GAC), as well as domain V helices H80-H88 (Supplementary Fig. 4d, f, m and Supplementary Fig. 8c), representing additional examples for rRNA induced rRNA stabilization or folding. However, the CP remains mobile and upon rotational

movement of the 5S rRNA towards the subunit interface, H84 and H38 stabilize each other (Fig. 5k, l).

### Late 50S assembly

Late 50S assembly, apart from core-stabilizing incorporation of uL2 (Fig. 5m and Supplementary Fig. 7a, d–g), involves stable binding of bL9 and bL28 in state *C-CP_L28*, which stabilizes the base of the L1 stalk (Supplementary Fig. 4o and Supplementary Fig. 7h–i). Next, H68 can form upon integration of bL33 and bL35 (Supplementary Fig. 4q and Supplementary Fig. 7j–k), while bL35 rigidifies the CP (Fig. 5n). In addition, the fixation of H68 seems to involve contacts with uL1. Furthermore, the L1-stalk associated H75 and CP associated H88 provide RNA contacts and stabilize H68. Accordingly, late step 1 assembly terminates with the "48S" particle *C-CP-H68*, sharing most structural features with a mature 50S subunit, but lacking a properly arranged FC. It is remarkable that the evolutionary most ancient part assembles last, both in vivo[12] and in vitro[14]. The extensive lack of L-proteins within the FC rationalizes why numerous assembly factors (such as EngA, DbpA, RlmE, and ObgE[13,32–35]) are required to finalize it in vivo and why high thermal energy is required to activate the particle in vitro[14].

## Discussion

Here we show, using the 50S in vitro reconstitution assay in combination with cryo-EM and multi-particle refinement that already after 3 min of reaction as much as 16 structurally distinct precursors have formed. Hence, the structures we report provide unprecedented insights into the detailed progression of 50S in vitro assembly, addressing long-standing questions and revealing fundamental principles of large subunit assembly.

One important question related to in vitro ribosome assembly assays is, how subunit assembly can take place, despite the presence of full 16S or 23S rRNA? In principle, structure formation could initiate at multiple, possibly competing sites. While we cannot rule out this scenario (based on our qMS data), we did not obtain structural evidence for any nucleation point other than domain I (Supplementary Fig. 6). Instead, our data indicate that the assembly of the 50S subunit does not proceed arbitrarily but has its morphogenetic origin in domain I and overall follows a 5´ to 3´ direction.

Based on their intrinsic affinities, uL22, uL24, and uL29 interact with their RNA binding sites in domain I and nucleate 50S assembly. Consequently, binding sites for uL4 and bL23 are formed. Next, binding of uL4 and bL23 initiates formation of domains II and III, respectively, and further proteins join, exemplifying the inherent cooperativity of ribosome assembly. Hence, we visualize Nomura´s paradigm, that all the information for ribosome assembly is contained in the structure (i.e., the chemical affinities) of the participating components[36]. As a practical consequence, atomic models derived from the assembly intermediates will be useful for the design of single molecule approaches and pharmacological agents interfering with critical transitions along the 50S assembly pathway.

After formation of domain I, assembly continues uL4 dependent with *d1_L4* and branches either to route 1-2-6 or 1-3-6 (Fig. 6). The presence of domain II apparently facilitates docking of the CP-forming 5S RNP and as a consequence, route 1-2-6 involves the possibility of CP formation either on *d12*, or *d126* resulting in *d12-CP*, or *d126-CP*, respectively. However, completion of the CP can also occur after the formation of state *C* during incremental late assembly (Supplementary Fig. 5). Route 1-3-6 yields state *C* due to presence of domain III, which is seeded by uL23, and in turn facilitates the incorporation of domain VI independently of domain II (Fig. 6 and Supplementary Fig. 5). Furthermore, additional routes are conceivable. Precursor *d12_L23* could derive from both *d12* and *d1_L4/L23* and constitute an intermediate that matures to state *C* by completion of domain III and simultaneous or subsequent formation of domain VI. Proteins that contribute to the formation of these domains are bL17, bL32, and uL3 (Fig. 6).

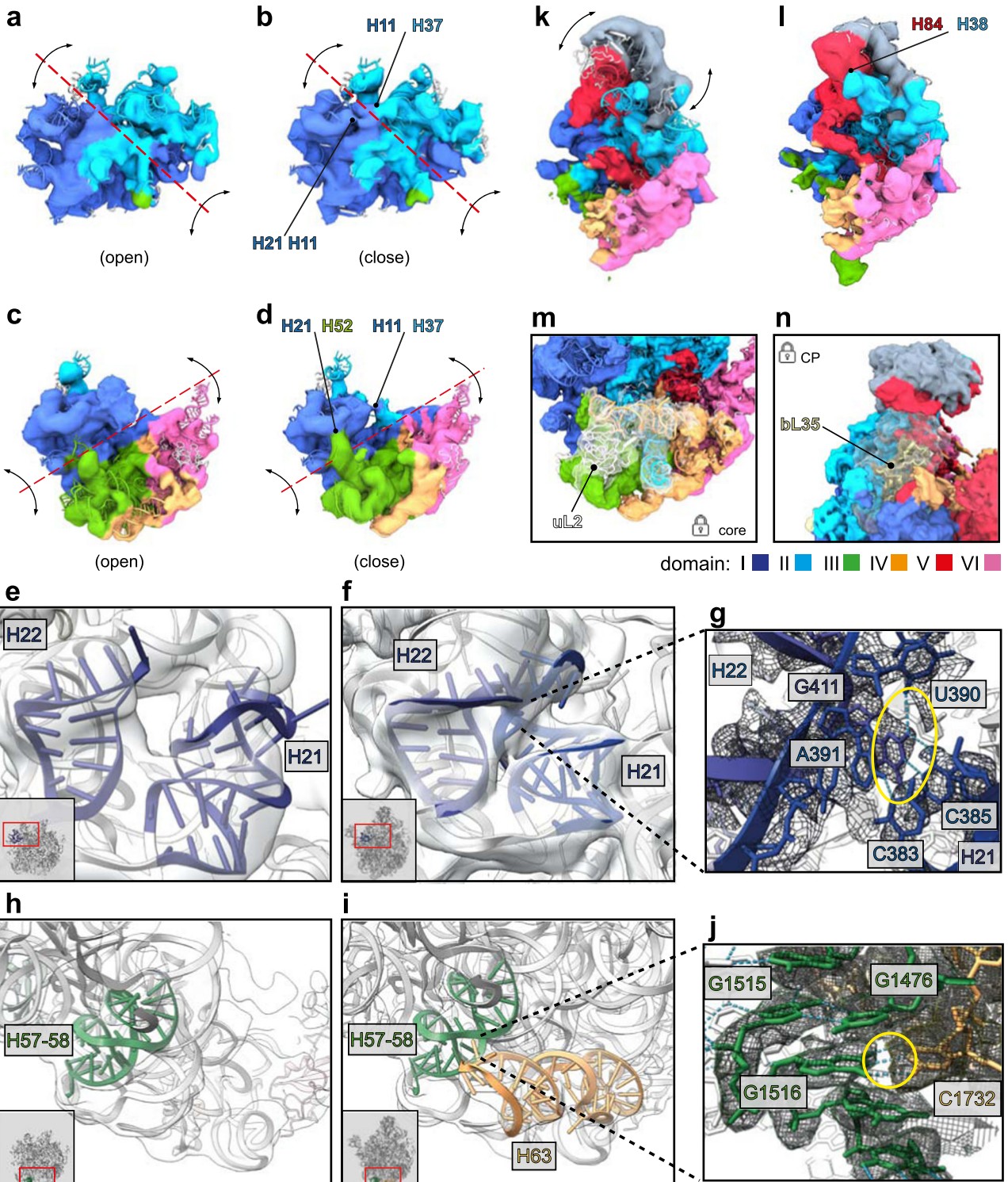

**Fig. 5 | Tertiary contacts between rRNA helices.** 3D variability maps of states *d12* (**a**, **b**), d136 (**c**, **d**), *d126-CP* (**k**, **l**). First (left) and last (right) reconstructed frame of each state is shown. Maps were filtered to 8 Å and color-coded according to the six architectural domains of 23S rRNA. Red dotted lines and black arrows indicate axis and directions of observed global movements, respectively. Stabilizing contacts in closed conformations are indicated by black arrows. Interacting regions are labeled. Cryo-EM maps (transparent gray) and PDB models corresponding to rRNA helices H21 and H22 (blue), H57-58 (green), or H63 (orange), respectively. Local snapshots of state *C⁻* (**e**, **h**) and state *C-CP_L2/L28* (**f**, **i**) contain thumbnails with red rectangles indicating the region of interest. **g**, **j** close-ups of regions of interest in state *C-CP_L2/L28*. Yellow circles highlight H-bond interactions (light blue dotted lines) between C385-G411 (**g**) and G1516-C1732 (**j**). **k**, **l** State d126-CP exhibiting flexibility in the CP occupying a more upright (**k**) or more inclined position towards the interface (**l**). Binding sites of L-proteins uL2 (state *C_L2*) (**m**) and bL35 (state *C-CP-H68*) (**n**).

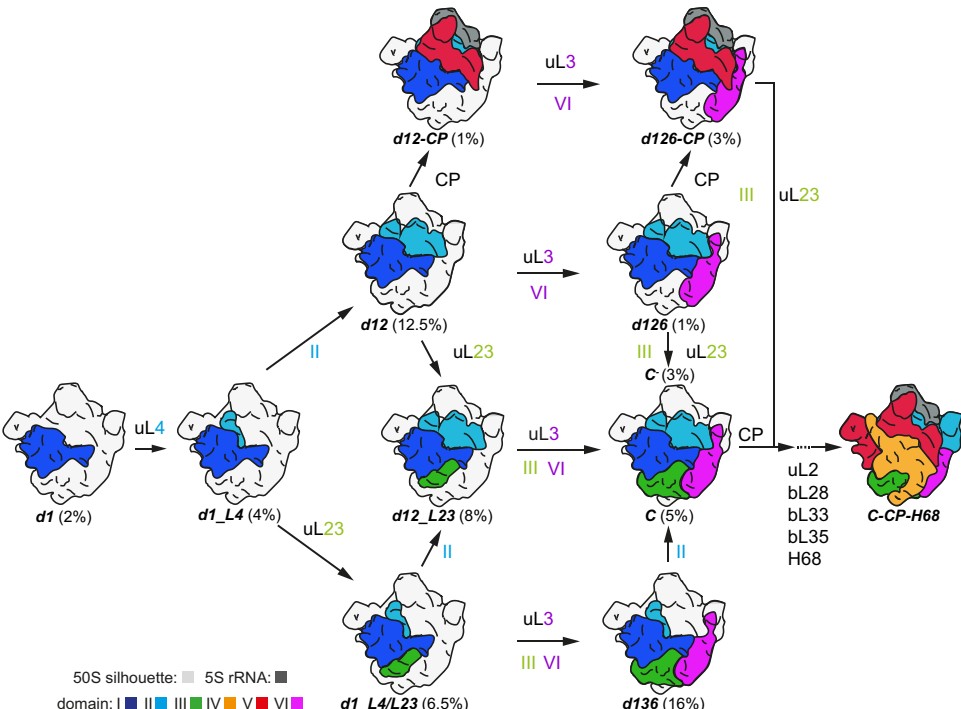

**Fig. 6 | Model of early 50S assembly.** Models of individual states color-coded as indicated. Numbers in % represent their relative abundance in the sorting scheme. Gray silhouette illustrates density of the mature 50S subunit. The labels uL4, uL23, uL3 and the numbers II, III, and VI indicate when cryo-EM densities for parts of these proteins or entire 23S rRNA domains, respectively, become detectable. CP central protuberance; H68 helix 68.

Accordingly, early 50S assembly is characterized by hierarchical arrangements of rRNA, resulting in branches, but on the other hand, generates cooperativity and ultimately culminates in convergence along the assembly pathway. Due to the interwoven architecture and the ring-shaped arrangement of domains I, II, III, and VI, domain VI can be incorporated via routes 1-2-6 or 1-3-6. While uL24 promotes the formation of the 5′-domain I, the late assembling uL3 plays a critical role in the structural organization of the 3′ end-containing domain VI. As we do not observe a state consisting only of domains I and VI, their contact areas may be too small to drive stable incorporation of domain VI in the absence of either domain II or III. This suggests that domain I together with domain VI cradle the missing domain in *d126*, or *d136*, to promote the appearance of domain III or II, respectively, to conclude core formation and early assembly (Fig. 6). Thus, our analysis indicates that coordinated structural formation of the two terminal domains of 23S rRNA is a decisive event that drives further assembly.

Structural data regarding earliest stages of large subunit assembly are lacking from both the prokaryotic and the eukaryotic domain. Interestingly, a structural analysis of nucleolar pre-60S assembly captured a state, consisting of 28S rRNA domains I and II (similar to our state *d12*), that was thought to proceed linearly to a 60S subunit[26]. Surprisingly, a second study found evidence for parallel assembly pathways[27]. While we find similar states, confirming the evolutionary conservation of the process, our analysis starts with a particle whose assembly has just initiated (*d1*). In addition, we can take advantage of the multitude of states we were able to obtain after thorough sorting and trace how decisive steps are taken towards a mature 50S subunit, involving multiple parallel routes of assembly.

We took advantage of the fact that the in vitro reconstitution occurs in the absence of ribosome assembly factors. Hence, precursor states approaching at rate limiting steps of the reaction are not as rapidly processed as is the case in vivo and in in vitro systems utilizing assembly factors. Dong et al.[37] used a system to study 50S assembly that involves in vitro transcribed ribosomal rRNA (iSAT) and combined it with a cryo-EM analysis. Their setup is supposed to be closer to the in vivo situation, since in addition to vectorially produced rRNA, it also utilizes S150 extract containing all relevant ribosome assembly factors. Interestingly their analysis yielded roughly the same structural states, suggesting that the states we obtained are of physiological relevance. As our system enriches certain states (such as *d12*, *d12_L23*, and *d136*), we have been able to refine highly resolved maps of these interesting early pre-50S intermediates.

In addition, we noticed that the assembly pathways that we describe structurally are in very good agreement with the Nierhaus assembly map of the 50S subunit[38] (Supplementary Fig. 5b).

Taken together, the combination of time-limited in vitro 50S assembly and cryo-EM allowed us to obtain unprecedented insights into the early assembly phase of the bacterial large ribosomal subunit. However, to obtain a more detailed understanding of the assembly process, fluorescence-based real time experiments and approaches clarifying the mechanistic role of ribosome assembly factors will be required.

## Methods

### 50S in vitro reconstitution

The assay was performed according to[39] with the modifications described in[14] as follows. For the step 1 reaction of the 50S reconstitution, seven samples were prepared in individual tubes corresponding to 7 time points. In each of the tubes, 6.25 equivalent units of TP50 (total protein of the 50S subunit derived from *E. coli* CAN20-12E (RNase I⁻ II⁻ D⁻ BN⁻)[40]) and 6.25 $A_{260}$ rRNA (purified 23S rRNA + 5S rRNA) were mixed on ice and incubated at 44 °C using a thermo-block (Eppendorf, Thermomixer comfort) in Rec4 buffer (20 mM HEPES/KOH pH 7.6, 400 mM NH₄Ac, 4 mM Mg(Ac)₂, 0.2 mM EDTA, 5 mM 2-mercaptoethanol). A first aliquot was taken instantaneously (0 min), and further samples were taken after 2, 3, 5, 10, 20, and 30 min of incubation. All reactions were stopped by placing the individual tubes on ice. For an activity assay (in vitro translation assay), 36 pmol of material from the individual time points were adjusted to 20 mM Mg(Ac)₂ concentration and incubated for 90 min at 50 °C.

## Analytical sucrose density gradient ultra-centrifugation

18 pmol material from the individual time course were subjected to analytical sucrose density gradient (10–30%) ultra-centrifugation (SW40 rotor, 26,000 rpm (-85,000 g) for 17 h) in Tico buffer (20 mM HEPES/KOH pH7.6 on ice, 30 mM KAc, 6 mM Mg(Ac)$_2$, 4 mM 2-mercaptoethanol).

## In vitro translation assay (poly(U)-dependent poly(Phe) assay)

6 pmol isolated native 50S subunits (50S), reconstituted 50S subunits (50S rec) from each time point of step 1 that had been subjected to a full step 2 incubation, were mixed with 12 pmol 30S subunits and incubated in assay buffer (20 mM HEPES/KOH pH7.6, 150 mM NH$_4$Ac, 6 mM Mg(Ac)$_2$, 0.05 mM EDTA, 4 mM 2-mercaptoethanol, 0.05 mM spermine, 2 mM spermidine, 3 mM ATP, 1.5 mM GTP, 5 mM acetyl-phosphate, 83 μM [$^{14}$C] Phe (22 dpm/pmol) (Hartmann analytic GmbH), 0.83 mg/ml poly(U), 0.34 mg/ml tRNA$^{bulk}$ and S100 extract) for 60 min at 37 °C. Subsequently, 30 μl BSA (1% stock solution) and 2 ml TCA (10%) were added. Samples were briefly vortexed, incubated for 15 min at 90 °C and 5 min on ice. Precipitated proteins were immobilized on glass filters, washed two times with 5% TCA and once with ether/ethanol (1:1). The filters were incubated in scintillation liquid (Roth, Rotiszint eco plus) and the amount of incorporated $^{14}$C Phe was determined using a scintillation counter (Wallac 1409 liquid scintillation counter). The assay was performed in duplicates and mean values were calculated.

## Quantitative mass spectrometry analysis

108 pmol of the individual samples from the step 1 time course reaction were adjusted to total volume 180 μl and loaded on 900 μl sucrose cushion (20% in Tico buffer). Ultra-centrifugation was performed (TLA-110 rotor, 65,000 rpm (-151,000 g) for 2.5 h). After removing the supernatant, the pellet was briefly washed and dissolved in Tico buffer. 36 pmol of each sample were analyzed by quantitative mass spectrometry according to the workflow described in ref. [13]. For protein comparisons per sample, iBAQ intensities were normalized to uL24, which is an early assembly protein of the large bacterial ribosomal subunit. Fold changes to the reference purified mature 50S was calculated from normalized iBAQ values. The ribosomal stoichiometry of individual L-proteins in each sample was calculated as the percentual fraction of the value of corresponding L-proteins in mature 50S. Therefore, 100% stoichiometry represents a protein that is as abundant as its counterpart in the mature 50S subunit. Experiments were performed in duplicates and data were processed with Heatmap Illustrator 1.0 (CUCKOO Workgroup) and Origin8 (OriginLab Corporation).

## Sample preparation for cryo-EM

12 pmol of the 3 min sample purified via sucrose cushion ultra-centrifugation, as described above, were diluted twofold in adaptation buffer (20 mM HEPES/KOH pH7.6, 4 mM Mg(Ac)$_2$, 0.2 mM EDTA, 5 mM 2-mercaptoethanol) to achieve a final concentration of 288 pmol/ml and spotted on glow-discharged holey carbon grids (R1.2/1.3 copper 400 mesh holey carbon, without additional carbon film on top (carbon-free)), (Quantifoil Micro Tools GmbH) and cryo plunged in liquid ethane after blotting using a Vitrobot device (MK4). For structural analyses of ribosome assembly, we generally use grids without thin carbon to minimize a preferred orientation of the particles.

## Cryo-electron microscopy and data processing

Data were collected on a FEI TecnaiG2 Polara equipped with a Gatan K2 Summit detector operated in super-resolution mode at 300 kV at a calibrated pixel size of 0.625 Å. Movies were acquired for 10 s applying a total electron dose of 63 e$^-$/Å$^2$.

## Data processing

Movies were aligned and dose-weighted using Defocus values were estimated using Gctf[41]. Templates for particle picking were generated in SPIDER[42]. Therefore, density maps were generated from atomic models of state 1 (PDB: 6GC7) and state 5 (6GBZ), subsequently low pass filtered to 20 Å, followed by projection into 84 equally distributed orientations, respectively. Orientation images were averaged into four projections using Xmipp3 2D classification[43] classification for each template, resulting in eight distorted averages (diameter: 140–230 Å) for particle picking. Particles were picked using templates in Gautomatch (developed by K. Zhang). Particle images were extracted and normalized, using Relion 3.0[44] with a box size of 600 and Fourier cropped to 150 for sorting. If not stated otherwise, Cryosparc v3.1[45] was used for the identification and refinement of final classes. Initial sorting was achieved using ab-initio classification, followed by two rounds of heterogeneous refinement to recover ribosomal particles (Supplementary Fig. 2a). Remarkably, recovered ribosomal particle classes exhibited density for the architectural domains 1 and 2 of the 23S rRNA only in the first, and density for domain 1 only in the second heterogeneous refinement. Selected particles were then pooled and aligned to a consensus map using Homogenous refinement. Remaining non-ribosomal particles were sorted out using reference-free 2D classification. Initial identification of different 50S precursor classes was achieved using 3D classification without alignment in Relion 3.1. Therein, best classification results were achieved, when particles were locally aligned to domain 1 of the 23S rRNA using Local refinement in prior. The 3D classification resulted in six distinct classes (d1, d12, d136, C, d126-CP, C-CP). All particles were re-aligned and re-assigned using identified classes as templates and heterogeneous refinement (Supplementary Fig. 2b). An additional 3D ice template was generated as non-ribosomal bait template by ab-initio reconstruction of respective classes from the 2D classification. Re-assigned particles were refined and remaining structural heterogeneity was sorted using hierarchical 3D variability clustering. When necessary, additional rounds of multi-class ab-initio reconstruction (class similarity = 0) were performed, and non-structured maps were used as bait templates in one round of heterogenous refinement to sort out shiny particles. The most mature precursor, C-CP-H68, still showed marked fragmented density for uL16, bL35, and H68, indicating remaining structural heterogeneity. However, due to low particle numbers, it was not sorted further. After completion of particle classifications, sub-classes were refined using non-uniform refinement at a pixel size of 1.25 Å.

## Model building

Atomic models of previously identified 50S assembly intermediates (PDB: 6GC0, 6GC4, 6GC7, 6GC8) were used for modeling. Initially, models were rigid body docked in ChimeraX[46], followed by rigid body fitting of individual L-proteins. RNA regions and L-proteins were removed when densities were missing or strongly fragmented. The adjustment of structured elements was performed by iterative model building and real-space refinement into the EM-density using Coot 0.9.6[47], Phenix 1.19[48] and ERRASER[49] considering secondary structure restraints. Models for maps with resolutions above 4 Å were subjected to a final round of geometry minimization in Phenix for restoration of geometry of the RNA. For the analysis of L-protein mediated sampling (Fig. 3), full-length models of respective L-proteins were fitted into EM maps of indicated states. A 50S Cryo-EM map (EMD-222614) was used for comparison[50]. Models were colored according to any rRNA contacts the corresponding 23S architectural domains within 4 Å distance. Self-contacts and surface exposure were colored in gray. Local Cryo-EM maps are shown at 3 Å distance to atomic models using the ChimeraX zone tool. For better visualization, noise and small fragmented regions were excluded by using the surface dust tool.

## Reporting summary

Further information on research design is available in the Nature Portfolio Reporting Summary linked to this article.

## Data availability

Cryo-EM density maps and atomic models are stored in EMDB and PDB as follows: *d1* (EMD-16509/ PDB ID 8C9C), *d1_L4* (EMD-16508/ PDB ID 8C9B), *d1_L4/L23* (EMD-16507/ PDB ID 8C9A), *d12* (EMD-16506/ PDB ID 8C99), *d12_L23* (EMD-16505/ PDB ID 8C98), *d136* (EMD-16504/ PDB ID 8C97), *d126* (EMD-16503/ PDB ID 8C96), *d12-CP* (EMD-16502/ PDB ID 8C95), *d126-CP* (EMD-16501/ PDB ID 8C94), *C* (EMD-16500/ PDB ID 8C93), *C* (EMD-16499/ PDB ID 8C92), *C_L2* (EMD-16498/ PDB ID 8C91), *C-CP* (EMD-16497/ PDB ID 8C90), *C-CP_L28* (EMD-16496/ PDB ID 8C8Z), *C-CP_L28/L2* (EMD-16495/ PDB ID 8C8Y), *C-CP-H68:* (EMD-16494/ PDB ID 8C8X). Mass spectrometry proteomics data have been deposited to ProteomeXchange Consortium (http://proteomecentral.proteomexchange.org) via the PRIDE partner repository with the dataset identifier PXD030312. All additional data needed to evaluate the conclusions in the paper are present in the paper and/or the Supplementary Materials. Source data are provided with this paper.

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

## Acknowledgements

We would like to thank Anett Unbehaun, Timo Fluegel and Matthew Kraushar for intensive scientific discussions, excellent suggestions, and text editing. This work was funded by Bundesministerium für Bildung und Forschung (BMBF 16GW0300 to C.M.T.S.) and the Human Frontier Science Program Organization (HFSP-Ref. RPG0008/2014, to C.M.T.S.). In addition, this work was supported by the Deutsche Forschungsgemeinschaft (DFG) through the cluster of excellence Unifying Systems in Catalysis (UniSysCat) under Germany´s Excellence Strategy-EXC 2008-390540038 (to A.B., C.M.T.S., and P.S.).

## Author contributions

B.Q. performed all biochemical experiments and prepared sample for cryo-EM and qMS; S.M.L. processed, analyzed, and visualized structural data and built atomic models. A.B. and P.S. evaluated and improved atomic models; C.H.V.-V. and M.S. performed qMS and analyzed the data. J.B. and T.M. participated in the grid preparation, operation of the microscopes, data acquisition, and processing; C.M.T.S. and R.N. supervised the study. R.N. drafted the manuscript. All authors contributed to writing of the manuscript.

## Funding

## Competing interests

The authors declare no competing interests.
