## [Peer Review File · Nature Communications]

Cryo-EM captures early ribosome assembly in actionREVIEWERS' COMMENTS

Reviewer #1 (Remarks to the Author):

In this work, Qin et al. detail a series of cryo-EM structures that help the field visualize the large ribosomal subunit undergoing some of the earliest stages of assembly in vitro. Most critically, they determine a low-resolution structure of the earliest assembly particle yet observed, which is composed of a single rRNA domain bound to 3 r-proteins. Further, from this same sample, they resolve 15 additional mature structures and make good progress in rationally ordering these structures into a plausible assembly pathway. In doing so, they make interesting observations about how cooperativity manifests during assembly through specific coupled rRNA folding and r-protein binding events. Overall, the work is conducted with rigor, the conclusions are clearly described, and the results – specifically the early assembly intermediate particles – will be of great interest to those studying ribosome biogenesis. Overall, I support the publication of this work after the authors have considered the following concerns.

1) This work does not describe “time-resolved cryoEM” – a single timepoint from a reaction was analyzed and all of the proposed sequential models were derived from this single snapshot. Leaving ‘time-resolved’ in the title may mislead readers into thinking multiple time-dependent samples were analyzed, and that the resulting assembly models were derived from time-dependent changes in relative populations of these states. Such a study would be interesting, but is not the one described. Note, I am not suggesting that the authors analyze additional time points by cryo-EM, but instead simply adjust the prose.

2) It is unclear how (or if) assembly reactions were quenched before adding them to the sucrose cushion to separate free/bound r-proteins for analysis by mass spectrometry. If no quench was added, assembly was likely to occur while loading gradients, during the early stages of the spin/etc. This could explain the nearly stoichiometric (>70%) occupancy in even the earliest timepoints. If no quench were added, this section should be rewritten to simply focus on the 3 min timepoint, and the time-dependent element of the MS analysis should be eliminated.

3) Detailed molecular morphogenesis – contributions of L-proteins. As I understand, in all three instances described, the author’s structures show either both rRNA element and r-protein missing, or both elements present, with no evidence of r-protein binding preceding stable/mature rRNA folding - consistent with highly cooperative r-protein binding/rRNA folding. As written, this section suggests that r-protein binding precedes rRNA folding and that the protein actively ‘remodels’ or ‘creates seeds’ for the rRNA to fold/dock. Given the cooperative nature of these events, such an ordering is plausible, but impossible to assess, and rRNA folding could just as plausibly occur spontaneously, thereby creating a high-affinity r-protein binding site that, once bound by r-protein, restricted spontaneous rRNA denaturation. The more active role of r-proteins and the implied kinetic ordering the authors described should be reframed with the above considerations in mind.

These concerns aside, this is a wonderful paper, and I congratulate the authors on their study.

Joey Davis

Reviewer #2 (Remarks to the Author):

This exciting manuscript from Spahn & Nikolay's groups provides important structural insights on the early stages of the bacterial 50S ribosomal subunit. The group uses a purely in vitro ribosome reconstitution system. Ribosome purified components are mixed, and samples are collected at different time points. Sucrose gradient centrifugation was used to determine that after three-minute incubation, the sample was 'richer' and contained a higher variety of assembly intermediates. Consequently, the authors picked the three-minute time point and analyzed it using cryo-EM. Extensive image classification allowed them to visualize 16 different ribosome intermediates. These structures were resolved to a sufficient resolution to inform mechanistic insights regarding how the assembly of the large ribosomal subunit occurs.

The most novel aspect of this study is that it describes for the first-time structures of early 50S assembly intermediates. Most of the existing cryo-EM studies in 50S subunit biogenesis have extensively characterized late assembly intermediates. Before this study, researchers assumed earlier 50S intermediates were highly flexible and not amenable to cryo-EM single particle techniques because these methods rely on averaging of particle images of structurally congruent intermediates.

Another interesting aspect of the study is that even though it was well known that r-proteins promote the folding of rRNA, the provided structures in this manuscript allow visualizing this aspect of their functionality directly. For example, they show that uL4 and uL23 contribute to the formation of local seeds and rRNA helices to promote further assembly of domains II and III and determine whether assembly proceeds along one of the various possible parallel pathways. (1-3-1 or 1-2-1).

Other aspects of the study are less novel but are consistent with previous work in the field and provide reassurance of some other well-known aspects of the 50S subunit assembly using a purely in vitro system in which assembly is studied in the complete absence of ribosome assembly factors. This includes the observation that the assembly of the 50S subunit can proceed following multiple parallel pathways. Also, a possible and preferred path of assembly involves the assembly of the core particle first, followed by the assembly of the CP, L1 and L7/12 stalks. The functional core is always assembled at the latest stages, and indeed it does not occur without step 2 of the in vitro reconstitution reaction

involving heating the sample at 50 C. This is consistent with the already known fact that assembly factors are essential for the maturation of the structural elements comprising the functional core.

Given that 50S assembly is assayed in the complete absence of assembly factors, this study describes the mechanisms of assembly as they are 'wired' in the ribosome components (23S rRNA and r-proteins). Based on the observed structures, an exciting aspect is that the 23S rRNA tend to fold following an overall 5' to 3' direction. In vivo, this folding order is favored because of the simultaneous rRNA transcription and folding. However, I found it remarkable that the 23S rRNA can behave similarly under in vitro conditions where the entire 23S rRNA molecule is available from the onset.

Due to the fact that the in vitro assembly reaction does not contain any assembly factors, the authors were able to visualize a few assembly intermediates that have not been seen in vivo previously. These include assembling states such as d12-CP and d126-CP. In both states, the CP assembles before the core of the particle is completed. These structures validate the already known fact that assembly occurs following parallel pathways and that the role of the assembly factors in vivo is to channel the particles down to a few of the possible parallel pathways of assembly. This is one aspect that is missing from the discussion. Observations are described and discussed in the context of the assembly of the 50S subunit. However, it should be made clear that the assembly process described here is the one 'wired' within the 50S subunit components. It is possible that some of the observed states, even though they are thermodynamically possible, they are never seen in the in vivo context due to the presence of assembling factors providing directionality to the process.

Overall, I found the paper well written, but sections in the results and methods read cryptic. Because figures and extended figures are well designed and straightforward, it is possible to finally understand the descriptions of the results section of the manuscript. I think the entire results section of the manuscript would benefit from a bit more extended description of the data, figures and results. Nature Communications is a general audience journal, and this improvement would help readers interested in macromolecular assembly processes but unfamiliar with the ribosome assembly process.

Given the new early assembly intermediates described for the first time in this study, it would have been helpful to include an analysis of how the r-proteins observed in these assembly intermediate and their entry order agree or disagrees with the Nierhaus 50S assembly map. This type of analysis was done in a previous study from this group describing later assembly intermediates (Nikolay et al. (2018), Molecular Cell 70, 881–893; Figure S7). It would be a nice addition to the present manuscript.

A critical issue to solve is the description of the image processing workflow. In particular, the approach followed to sort out the different classes is cryptic and unclear. I cannot match the text in the materials and methods section describing the image classification with the extended figure 2 illustrating the classification approach. I would advise the authors to clarify the text and extended figure 2. The reader

needs to understand this aspect of data processing since it determines the assembly intermediates obtained in the assembly reaction and from where mechanistic insights about ribosomal assembly are withdrawn. To be more precise, I can follow extended figure 2 panel A, but I am unclear on how the nine classes in panel A (right side) relate to classes in panel B. What does it mean AI + HR clean up? I presume HR is heterogeneous refinement, but I cannot figure out what AI is. The text on the figure legend or description of the methods does not help either. The description of the classification must be very clear, as most of the conclusions in this manuscript rely on the correctness of this data processing.

It will also be necessary the authors to clarify why particle picking was done using template matching. Using templates for particle picking is appropriate when there is some a priori knowledge of the assemblies in the micrographs. This is the case here for late assembly intermediates. There are assembly intermediates present in this dataset similar to the state 1 and 5 used by the authors from their previous study in late 50S assembly intermediates. Using these templates bias the particle selection process to pick assembly intermediates from the mid to late stages of assembly. The aim and novelty of this study sit mainly with the description of early assembly intermediates for which there were no a priori structural information. I would argue that the authors should have started the picking process with a 'blob-picker' approach or, even better, using Topaz or a deep-learning particle-picking approach. Such approaches are most likely to achieve a less biased particle selection and ensure that relevant early assembly intermediates, which are significantly different from the mature 50S subunit, are not left behind in the micrographs. The most significant novelty in this manuscript is the description of these early assembly intermediates. How can the authors be sure they capture all the early intermediates present in the micrographs? There may be many other classes in addition to those described, and the missing classes could have changed the study's overall conclusion.

It has been described that EM grids without additional carbon film expose ribosomal particles (or any sample) more intensively to the air-water interface. This is not an issue with mature ribosomal particles because r-proteins are tightly bound. However, this may represent an issue with assembly intermediates where some of the r-proteins are not tightly bound and could lead to the loss of some of the r-proteins present in the intermediate upon repeated interaction with the air-water interface. How do the authors rule out this possibility?

Other suggested changes/corrections:

It seems figure legend for figure 5, the panel description does not correspond to the panels.

Panel J is always skipped in figures with panels named A to K or more. Is there any reason not to call the panels strictly alphabetical? Fig 2 and Fig 5 are two examples.

Figure 3 is difficult to figure out what part of the 50S subunit we are looking at in each panel and what the view direction is. Some guidance panels, such as those used in other figures, would be helpful to make this figure easier to understand.

REVIEWERS' COMMENTS

Reviewer #1 (Remarks to the Author):

In this work, Qin et al. detail a series of cryo-EM structures that help the field visualize the large ribosomal subunit undergoing some of the earliest stages of assembly in vitro. Most critically, they determine a low-resolution structure of the earliest assembly particle yet observed, which is composed of a single rRNA domain bound to 3 r-proteins. Further, from this same sample, they resolve 15 additional mature structures and make good progress in rationally ordering these structures into a plausible assembly pathway. In doing so, they make interesting observations about how cooperativity manifests during assembly through specific coupled rRNA folding and r-protein binding events. Overall, the work is conducted with rigor, the conclusions are clearly described, and the results – specifically the early assembly intermediate particles – will be of great interest to those studying ribosome biogenesis. Overall, I support the publication of this work after the authors have considered the following concerns.

> We are grateful for the positive feedback from reviewer #1 and do our best to address his concerns.

1) This work does not describe “time-resolved cryoEM” – a single timepoint from a reaction was analyzed and all of the proposed sequential models were derived from this single snapshot. Leaving ‘time-resolved’ in the title may mislead readers into thinking multiple time-dependent samples were analyzed, and that the resulting assembly models were derived from time-dependent changes in relative populations of these states. Such a study would be interesting, but is not the one described. Note, I am not suggesting that the authors analyze additional time points by cryo-EM, but instead simply adjust the prose.

> We thank the reviewer for pointing out that the title is misleading. Accordingly, we adjusted our title to “Cryo-EM captures early ribosome assembly in action”.

2) It is unclear how (or if) assembly reactions were quenched before adding them to the sucrose cushion to separate free/bound r-proteins for analysis by mass spectrometry. If no quench was added, assembly was likely to occur while loading gradients, during the early stages of the spin/etc. This could explain the nearly stoichiometric (>70%) occupancy in even the earliest timepoints. If no quench were added, this section should be rewritten to simply focus on the 3 min timepoint, and the time-dependent element of the MS analysis should be eliminated.

> Structural conversions in the precursors are known to require high amounts of (thermal) energy (44-50°C). We do not expect that exposure of the reaction to 0°C (on ice) and 4°C (in the ultracentrifuge) will permit structural transitions, neither in the short presence of TP50 prior to centrifugation, nor thereafter. Nevertheless, we rewrote the corresponding sections and clarified that “All reactions were stopped by placing the individual tubes on ice.” (line 302)

And: “12 pmol of the **3min sample** purified via sucrose cushion ultra-centrifugation, as described above, were diluted twofold in adaptation buffer...” (line 343)

3) Detailed molecular morphogenesis – contributions of L-proteins. As I understand, in all three instances described, the author’s structures show either both rRNA element and r-protein missing, or both elements present, with no evidence of r-protein binding preceding stable/mature rRNA folding - consistent with highly cooperative r-protein binding/rRNA folding. As written, this section suggests that r-protein binding precedes rRNA folding and that the protein actively ‘remodels’ or ‘creates seeds’ for the rRNA to fold/dock. Given the cooperative nature of these events, such an ordering is plausible, but impossible to assess, and rRNA folding could just as plausibly occur spontaneously, thereby creating a high-affinity r-protein binding site that, once bound by r-protein, restricted spontaneous rRNA denaturation. The more active role of r-proteins and the implied kinetic ordering the authors described should be reframed with the above considerations in mind.

> We thank reviewer #1 for pointing out these important caveats. We now address this issue specifically in lines 99-106.

“Whenever in the following we describe the presence, or appearance of an L-protein, an rRNA element, or an rRNA domain, it refers to a detectable, unambiguous density of this component in the corresponding cryo-EM map. Appearance of such densities can result from a component just binding at that moment or being part of the complex already, and structural rearrangements led to a more stable conformation capable of being captured by cryo-EM based technologies. This caveat is of particular importance in cases where densities for proteins and RNA elements appear simultaneously.”

These concerns aside, this is a wonderful paper, and I congratulate the authors on their study.
Joey Davis

> We thank reviewer #1 for his constructive and helpful criticism.

Reviewer #2 (Remarks to the Author):

This exciting manuscript from Spahn & Nikolay's groups provides important structural insights on the early stages of the bacterial 50S ribosomal subunit. The group uses a purely in vitro ribosome reconstitution system. Ribosome purified components are mixed, and samples are collected at different time points. Sucrose gradient centrifugation was used to determine that after three-minute incubation, the sample was 'richer' and contained a higher variety of assembly intermediates. Consequently, the authors picked the three-minute time point and analyzed it using cryo-EM. Extensive image classification allowed them to visualize 16 different ribosome intermediates. These structures were resolved to a sufficient resolution to inform mechanistic insights regarding how the assembly of the large ribosomal subunit occurs.

> We are pleased to hear that reviewer #2 appreciates the quality of our cryo-EM maps and supports our mechanistic conclusions.

1) The most novel aspect of this study is that it describes for the first-time structures of early 50S assembly intermediates. Most of the existing cryo-EM studies in 50S subunit biogenesis have extensively characterized late assembly intermediates. Before this study, researchers assumed earlier 50S intermediates were highly flexible and not amenable to cryo-EM single particle techniques because these methods rely on averaging of particle images of structurally congruent intermediates.

> In deed, we were also amazed that it was possible to reconstruct cryo-EM maps and build models for such early LSU assembly intermediates.

2) Another interesting aspect of the study is that even though it was well known that r-proteins promote the folding of rRNA, the provided structures in this manuscript allow visualizing this aspect of their functionality directly. For example, they show that uL4 and uL23 contribute to the formation of local seeds and rRNA helices to promote further assembly of domains II and III and determine whether assembly proceeds along one of the various possible parallel pathways. (1-3-1 or 1-2-1).

> Standing on the shoulders of giants it was a very exciting and satisfying discovery that we were able to provide structural evidence for what Dr. Nomura and others have predicted based on mutational and biochemical studies. We already dedicated a section of the discussion to this particular topic (lines 234-42).

3) Other aspects of the study are less novel but are consistent with previous work in the field and provide reassurance of some other well-known aspects of the 50S subunit assembly using a purely in vitro system in which assembly is studied in the complete absence of ribosome assembly factors. This includes the observation that the assembly of the 50S subunit can proceed following multiple parallel pathways. Also, a possible and preferred path of assembly involves the assembly of the core particle first, followed by the assembly of the CP, L1 and L7/12 stalks. The functional core is always assembled at the latest stages, and indeed it does not occur without step 2 of the in vitro reconstitution reaction involving heating the sample at 50 C. This is consistent with the already

known fact that assembly factors are essential for the maturation of the structural elements comprising the functional core.

> That is a very exciting point. We now refer to our previous study (Nikolay et al., 2018) and reemphasize the importance of assembly factors for the final maturation of the functional core that almost exclusively consists of rRNA (lines 216-20).

“It is remarkable that the evolutionary most ancient part assembles last, both *in vivo* (Davis, 2016) and *in vitro* (Nikolay, 2018). The extensive lack of L-proteins within the FC rationalizes why numerous assembly factors (such as EngA, DbpA, RlmE and ObgE (Zhang, 2014; Nicol, 1995; Arai, 2015; Nikolay, 2021)) are required to finalize it *in vivo* and why high thermal energy is required to activate the particle *in vitro* (Nikolay, 2018).”

4) Given that 50S assembly is assayed in the complete absence of assembly factors, this study describes the mechanisms of assembly as they are ‘wired’ in the ribosome components (23S rRNA and r-proteins). Based on the observed structures, an exciting aspect is that the 23S rRNA tend to fold following an overall 5’ to 3’ direction. *In vivo*, this folding order is favored because of the simultaneous rRNA transcription and folding. However, I found it remarkable that the 23S rRNA can behave similarly under *in vitro* conditions where the entire 23S rRNA molecule is available from the onset.

> In fact, we were amazed by this finding, as well and emphasized it both in the results and discussion sections (lines 109-12; 227-33).

5) Due to the fact that the *in vitro* assembly reaction does not contain any assembly factors, the authors were able to visualize a few assembly intermediates that have not been seen *in vivo* previously. These include assembling states such as d12-CP and d126-CP. In both states, the CP assembles before the core of the particle is completed. These structures validate the already known fact that assembly occurs following parallel pathways and that the role of the assembly factors *in vivo* is to channel the particles down to a few of the possible parallel pathways of assembly. This is one aspect that is missing from the discussion. Observations are described and discussed in the context of the assembly of the 50S subunit. However, it should be made clear that the assembly process described here is the one ‘wired’ within the 50S subunit components. It is possible that some of the observed states, even though they are thermodynamically possible, they are never seen in the *in vivo* context due to the presence of assembling factors providing directionality to the process.

> We thank reviewer #2 for this important suggestion. Shortly after submission of our manuscript to Nature Communications a biorxiv preprint from the Williamson lab appeared (Dong et al., 2022, biorxiv) that studies 50S assembly using *in vitro* transcribed ribosomal rRNA (iSAT) in combination with cryo-EM analysis. Their setup is supposed to be closer to the *in vivo* situation, since in addition to vectorially produced rRNA, it also utilizes S150 extract containing all relevant ribosome assembly factors. Interestingly, it turns out that they obtain roughly the same structural states as we do. Only the distribution of the states is different, suggesting that all the states we obtain are of physiologic relevance, but due to rate-limiting steps that are encountered in the

absence of assembly factors certain states are enriched in our analysis. We added a small chapter to the discussion to elaborate on this interesting observation (lines 275-84).

“We took advantage of the fact that the *in vitro* reconstitution occurs in the absence of ribosome assembly factors. Hence, precursor states approaching at rate limiting steps of the reaction are not as rapidly processed as is the case *in vivo* and in *in vitro* systems utilizing assembly factors. Dong et al. (Dong et al. 2022, biorxiv) used a system to study 50S assembly that involves *in vitro* transcribed ribosomal rRNA (iSAT) and combined it with a cryo-EM analysis. Their setup is supposed to be closer to the *in vivo* situation, since in addition to vectorially produced rRNA, it also utilizes S150 extract containing all relevant ribosome assembly factors. Interestingly their analysis yielded roughly the same structural states, suggesting that the states we obtained are of physiological relevance. As our system enriches certain states (such as *d12*, *d12_L23* and *d136*), we have been able to refine highly resolved maps of these interesting early pre-50S intermediates.”

6) Overall, I found the paper well written, but sections in the results and methods read cryptic. Because figures and extended figures are well designed and straightforward, it is possible to finally understand the descriptions of the results section of the manuscript. I think the entire results section of the manuscript would benefit from a bit more extended description of the data, figures and results. Nature Communications is a general audience journal, and this improvement would help readers interested in macromolecular assembly processes but unfamiliar with the ribosome assembly process.

> We thank reviewer #2 for bringing up this point. We carefully analyzed our manuscript and expanded our descriptions. In particular, the sections added in response to reviewer’s questions and suggestions lead to a more detailed description of the data.

7) Given the new early assembly intermediates described for the first time in this study, it would have been helpful to include an analysis of how the r-proteins observed in these assembly intermediate and their entry order agree or disagrees with the Nierhaus 50S assembly map. This type of analysis was done in a previous study from this group describing later assembly intermediates (Nikolay et al. (2018), Molecular Cell 70, 881–893; Figure S7). It would be a nice addition to the present manuscript.

> We thank reviewer #2 for pointing out to the Nierhaus assembly map. We noticed that our Supplementary Fig.5b is a structural interpretation of the Nierhaus map based on our data. Now we point this out in the main text (lines 285-86).

“In addition, we noticed that the assembly pathways that we describe structurally are in very good agreement with the Nierhaus assembly map of the 50S subunit (Herold, 1987) (Supplementary Fig. 5b).”

8) A critical issue to solve is the description of the image processing workflow. In particular, the approach followed to sort out the different classes is cryptic and unclear. I cannot match the text in the materials and methods section describing the image classification with the extended figure

2 illustrating the classification approach. I would advise the authors to clarify the text and extended figure 2. The reader needs to understand this aspect of data processing since it determines the assembly intermediates obtained in the assembly reaction and from where mechanistic insights about ribosomal assembly are withdrawn. To be more precise, I can follow extended figure 2 panel A, but I am unclear on how the nine classes in panel A (right side) relate to classes in panel B. What does it mean AI + HR clean up? I presume HR is heterogeneous refinement, but I cannot figure out what AI is. The text on the figure legend or description of the methods does not help either. The description of the classification must be very clear, as most of the conclusions in this manuscript rely on the correctness of this data processing.

>We thank reviewer #2 for pointing out these ambiguities. We revised the sections *Data processing*, (lines 363-93) *Model building* (lines 396-409) and *Supplementary Fig. 2* (lines 627-41), and hope that the descriptions are clear now.

„Data processing

Movies were aligned and dose-weighted using Defocus values were estimated using Gctf⁴². Templates for particle picking were generated in SPIDER⁴³. Therefore, density maps were generated from atomic models of state 1 (PDB: 6GC7) and state 5 (6GBZ), subsequently low pass filtered to 20 Å, followed by projection into 84 equally distributed orientations, respectively. Orientation images were averaged into four projections using Xmipp3 2D classification⁴⁴ classification for each template, **resulting in eight distorted/deformed averages (diameter: 140-230 Å) for particle picking. Particles were picked using templates in Gautomatch (developed by K. Zhang).** Particle images were extracted and normalized, using Relion 3.0⁴⁵ with a box size of 600 and Fourier cropped to 150 for sorting. If not stated otherwise, Cryosparc v3.1⁴⁶ was used for the identification and refinement of final classes. Initial sorting was achieved using ab-initio classification, followed by two rounds of heterogeneous refinement to recover ribosomal particles (**Supplementary Fig. 2a**). Remarkably, recovered ribosomal particle classes exhibited density for **the architectural domains 1 and 2 of the 23S rRNA only** in the first, and density for domain 1 only in the second heterogeneous refinement. **Selected** particles were then pooled and aligned to a consensus map using **Homogenous refinement**. Remaining **non-ribosomal** particles were sorted out using reference-free 2D classification. **Initial identification of different 50S precursor classes was achieved using 3D classification without alignment in Relion 3.1.** Therein, best classification results were achieved, when particles were locally aligned to domain 1 of the 23S rRNA using Local refinement in prior. The 3D classification resulted in six distinct classes (*d1*, *d12*, *d136*, *C*, *d126-CP*, *C-CP*). Particles were re-aligned and re-assigned using identified classes as templates and heterogeneous refinement (**Supplementary Fig. 2b**). An additional 3D ice template was generated as **non-ribosomal bait template** by ab-initio reconstruction of **respective classes from the 2D classification**. **Re-assigned particles were refined and remaining structural heterogeneity was sorted using hierarchical 3D variability clustering.** When necessary, additional rounds of multi-class ab-initio reconstruction (class similarity = 0) were performed, and non-structured maps were **used as bait templates in one round of heterogenous refinement** to sort out shiny particles. The most mature precursor, *C-CP-H68*, still showed marked fragmented density for uL16, bL35 and H68, indicating remaining structural heterogeneity. However, due to low particle numbers, it was not sorted further. After completion of particle classifications, sub-classes were refined using non-uniform refinement at a pixel size of 1.25 Å.

Model building

Atomic models of previously identified 50S assembly intermediates (PDB: 6GC0, 6GC4, 6GC7, 6GC8) were used for modeling. Initially, models were rigid body docked in ChimeraX⁴⁷, followed by rigid body fitting of individual L-proteins. RNA regions and L-proteins were removed when densities were missing or strongly fragmented. The adjustment of structured elements was performed by iterative model building and real-space refinement into the EM-density using Coot 0.9.6⁴⁸, Phenix 1.19⁴⁹ and ERRASER⁵⁰ considering secondary structure restraints. **Models for maps with resolutions higher/above 4 Å were subjected to a final round of geometry minimization in Phenix for restoration of geometry of the RNA.** For the analysis of L-protein mediated sampling (**Fig. 3**), full-length models of respective L-proteins were fitted into EM maps of indicated states. A 50S Cryo-EM map (EMD-222614) was used for comparison⁵¹. Models were colored according to any rRNA contacts the corresponding 23S architectural domains within 4 Å distance. Self-contacts and surface exposure were colored in gray. Local Cryo-EM maps are shown at 3Å distance to atomic models using the ChimeraX zone tool. For better visualization, noise and small fragmented regions were excluded by using the surface dust tool.

Supplementary Fig. 2: Sorting scheme

a) Sorting of particles and identification of initial classes. Extracted particles were subjected to an ab-initio reconstruction (AIR), followed by two rounds of heterogeneous refinement (HR) to recover ribosomal particles. Selected particles were refined to a consensus map, and remaining ice particles were sorted out using 2D classification. Particles were aligned to the 23S rRNA domain 1 region using local refinement and further sorted using 3D classification, skipping alignment in Relion 3.1. **b)** Re-assignment of initial classes, cleaning, and identification of final classes. Particles were re-assigned and re-aligned to previously identified classes. Templates used for particle re-assignment are labeled in blue. An additional bait template was generated using an AIR from selected non-ribosomal 2D classes. Further structural heterogeneity was identified using hierarchical 3D variability clustering. For *d1* and *d136* classes, assigned particles were cleaned using AIR followed by HR to identify the most defined subsets. Therefore, non-structured classes from AIR were used as bait templates, together with the starting ribosomal map in HR. Final classes are labeled in green. Sorting was performed at a pixel size of 3.75 Å. Final classes were refined at a pixel size of 1.25 Å (Supplementary Fig. 3).“

9) It will also be necessary the authors to clarify why particle picking was done using template matching. Using templates for particle picking is appropriate when there is some a priori knowledge of the assemblies in the micrographs. This is the case here for late assembly intermediates. There are assembly intermediates present in this dataset similar to the state 1 and 5 used by the authors from their previous study in late 50S assembly intermediates. Using these templates bias the particle selection process to pick assembly intermediates from the mid to late stages of assembly. The aim and novelty of this study sit mainly with the description of early assembly intermediates for which there were no a priori structural information. I would argue that the authors should have started the picking process with a ‘blob-picker’ approach or, even better, using Topaz or a deep-learning particle-picking approach. Such approaches are most likely to achieve a less biased particle selection and ensure that relevant early assembly intermediates, which are significantly different from the mature 50S subunit, are not left behind in the

micrographs. The most significant novelty in this manuscript is the description of these early assembly intermediates. How can the authors be sure they capture all the early intermediates present in the micrographs? There may be many other classes in addition to those described, and the missing classes could have changed the study's overall conclusion.

> We thank reviewer #2 for bringing up this valuable aspect, which actually was investigated in the initial design of the processing scheme. Different particle picking parameters were tested, before proceeding to subsequent processing steps. We decided to use template-matching for the following reasons:

(1) Most picking software, such as Gautomatch, only support the selection of one blob size, which might limit the potential to cover the full spectrum of precursor states.

(2) Our templates were generated from 84 different viewing angle projections from low pass filtered state 1 and state 5 maps (168 views total), which were then collapsed into four averages (8 total), resulting in eight highly distorted templates of different size. These generated templates had diameters ranging from 230Å to 140Å and allowed for successful detection of to the earliest state (dI : 120-150 Å).

(3) Blob-picking and template matching resulted in similar particle numbers. However, while template-matching achieved better particle centering, blob-picking resulted in a higher number of particles.

(4) The cross-correlation threshold for template-matching was set very low (0.2) to include particles that do not match the template entirely. False-positive picks were accepted at that stage and were discarded in subsequent processing steps. As we do have a substantial number of states that deviate from the initial template size, we believe that our approach was successful.

However, as we appreciate the importance of this argument, we have re-evaluated the data set using the blob-picking tool in CryoSPARC, which allows for the generation of blobs of different sizes. We have picked particles using blobs of sizes between 230 and 140 Å. Blob-picking resulted in a higher number of particles, but more particles needed to be discarded because they were picked off-center (see above). After re-assignment of the particles to our previously identified states, their percentage distribution was comparable with our initial approach. In addition, none of the states had improved resolution compared to our approach. Taken together, it is unlikely that the use of blob-picking would change the study's overall conclusion.

10) It has been described that EM grids without additional carbon film expose ribosomal particles (or any sample) more intensively to the air-water interface. This is not an issue with mature ribosomal particles because r-proteins are tightly bound. However, this may represent an issue with assembly intermediates where some of the r-proteins are not tightly bound and could lead to the loss of some of the r-proteins present in the intermediate upon repeated interaction with the air-water interface. How do the authors rule out this possibility?

> We see the problem of potential loss of loosely associated proteins. However, our previous analyses taught us that use of a thin carbon layer provokes a preferred orientation of LSU precursors. This effect turned out to be more severe the more immature the particles are and

resulted in corrupt cryo-EM density maps. Our reasoning for not utilizing thin carbon was added to the manuscript (lines 348-49).

“For structural analyses of ribosome assembly, we generally use grids without thin carbon to minimize a preferred orientation of the particles. “

Other suggested changes/corrections:

11) It seems figure legend for figure 5, the panel description does not correspond to the panels.

> The figure legend for Fig. 5 has been mixed up during the evolution of the manuscript. We are grateful to reviewer #2 for pointing this out. We corrected the figure legend and changed lines 215-19 in the main text accordingly.

“Late 50S assembly, apart from core-stabilizing incorporation of uL2 (Fig. 5m), involves stable binding of bL9 and bL28 in state *C-CP_L28*, which stabilizes the L1 stalk (Supplementary Fig. 4o and Supplementary Fig. 7h-i). Next, H68 can form upon integration of bL33 and bL35 (Supplementary Fig. 4q and Supplementary Fig. 7j-k), while bL35 rigidifies the CP (Fig. 5n).”

12) Panel J is always skipped in figures with panels named A to K or more. Is there any reason not to call the panels strictly alphabetical? Fig 2 and Fig 5 are two examples.

> The letter j is now added to the figure panels. We corrected both figures and main text accordingly.

13) Figure 3 is difficult to figure out what part of the 50S subunit we are looking at in each panel and what the view direction is. Some guidance panels, such as those used in other figures, would be helpful to make this figure easier to understand.

> We now have added a thumbnail to Fig. 3c indicating the viewing perspective. In addition, we denote in the figure legend that Fig. 3c is presented in “back view”. We noticed that Fig. 2 would benefit from such indication, too. Hence, we added a thumbnail to Fig. 2a indicating the viewing perspective.

We thank both reviewers for their constructive criticism.